# Information-Theoretic Descriptors of Molecular States and Electronic Communications between Reactants

**DOI:** 10.3390/e22070749

**Published:** 2020-07-07

**Authors:** Roman F. Nalewajski

**Affiliations:** Department of Theoretical Chemistry, Jagiellonian University, Gronostajowa 2, 30-387 Cracow, Poland; nalewajs@chemia.uj.edu.pl

**Keywords:** acid-base systems, density matrices, electron communications, reactivity theory, resultant information descriptors, state continuity

## Abstract

The classical (modulus/probability) and nonclassical (phase/current) components of molecular states are reexamined and their information contributions are summarized. The state and information continuity relations are discussed and a nonclassical character of the resultant gradient information source is emphasized. The states of noninteracting and interacting subsystems in the model donor-acceptor reactive system are compared and configurations of the mutually-closed and -open equidensity orbitals are tackled. The density matrices for subsystems in reactive complexes are used to describe the entangled molecular fragments and electron communications in donor-acceptor systems which determine the entropic multiplicity and composition of chemical bonds between reactants.

## 1. Introduction

The information theory (IT) [1,2,3,4,5,6,7,8] of Fisher [1] and Shannon [3] has been successfully applied in an entropic interpretation of the molecular electronic structure [9,10,11]. Several information principles have been investigated [9,10,11,12,13,14,15,16] and pieces of molecular electron density attributed to atoms in molecules (AIM) have been approached [12,16,17,18,19,20], providing the IT basis for the intuitive stockholder division of Hirshfeld [21]. Patterns of entropic bond multiplicities have been extracted from electronic communications in molecules [9,10,11,22,23,24,25,26,27,28,29,30,31,32], information distributions in molecules have been explored [9,10,11,33,34], and the nonadditive Fisher (gradient) information [1,2,9,10,11,35,36] has been linked to electron localization function (ELF) [37,38,39] of density functional theory (DFT) [40,41,42,43,44,45]. This analysis has formulated the contragradience (CG) probe for localizing chemical bonds [9,10,11,46], while the orbital communication theory (OCT) of the chemical bond using the “cascade” propagations in molecular information systems has identified the bridge interactions between AIM [11,47,48,49,50,51,52], realized through intermediate orbitals.

The quantum electronic states of molecular systems and their dynamics are determined by the Schrödinger equation (SE). These (complex) wavefunctions are specified by their modulus and phase components, which generate the probability and current distributions of the system electrons. Such physical attributes respectively reflect the complementary classical (static) and nonclassical (dynamic) structures of “being” and “becoming”, which both contribute to the state overall entropy and information content. It is of interest to examine their continuity relations in order to establish the net productions of these properties and to identify the origins of their sources. 

In quantum mechanics (QM), the wavefunction phase, or its gradient determining the effective velocity of probability density, give rise to nonclassical information and entropy supplements to the classical measures of Fisher [1] and Shannon [3]. In the resultant IT descriptors of electronic states, the information and entropy content in the probability (wavefunction modulus) distribution is combined with the relevant complement due to the current density (wavefunction phase) [53,54,55,56,57,58,59,60,61,62]. The overall (resultant) gradient information is then proportional to the expectation value of the system kinetic energy of electrons. Such combined descriptors are also required in the phase distinction between the bonded (entangled) and nonbonded (disentangled) states of molecular subsystems, for example, the substrate fragments of reactive systems [63,64,65]. This generalized treatment allows one to interpret the variational principle for electronic energy as equivalent information rule, and to use the molecular virial theorem [66] in general reactivity considerations [67,68,69,70,71]. The elementary chemical processes have also been monitored using the classical entropy and information descriptors [72,73,74,75]. 

Elsewhere, the application of DFT construction by Harriman, Zumbach, and Maschke (HZM) [76,77], of wavefunctions yielding the prescribed electron distribution, for a description of reactive systems has been examined [63,78,79]. In such density-constrained Slater determinants the defining equidensity orbitals (EO) of the Macke/Gilbert [80,81] type exhibit the same molecular probability density, with the orbital orthogonality being assured by the local phases alone. Such orbital configurations define a constrained multicomponent system, which is composed of the mutually-closed (disentangled) orbital units, with each subsystem being characterized by its own phase and chemical potential descriptors. Their simultaneous opening onto a common electron reservoir, and hence also onto themselves, generates an externally- and mutually-open orbital system, in which the EO fragments are effectively “bonded” (entangled) [63,82]. They, then, exhibit a common (molecular) phase descriptor and equalize their chemical potentials at the global reservoir level.

In OCT, the chemical-bond pattern of molecular systems can be probed [9,10,11,22,23,24,25,26,27,28,29,30,31,32] using techniques and descriptors developed in IT of communication devices (information channels) [3,4,7,8]. The conditional probabilities of simultaneous orbital events can be tackled using amplitudes from the bond-projected superposition principle (SP) of QM. Their cascade (“bridge”) propagations in molecules, involving intermediate orbitals, also determine the bridge contributions to chemical interactions between bonded atoms. This quantum approach to electronic communications defines the probabilities of observing specific orbital “events” conditional on the given “molecular” state, of the system as a whole, which can be used to generate the entire network of the state quantum communications between atomic orbitals or general basis functions. In the single Slater determinant approximation of the familiar Hartree–Fock (HF) or Kohn–Sham (KS) methods the communication amplitudes are related to the corresponding elements of the familiar charge-and-bond-order (CBO) matrix of quantum chemistry.

This novel IT framework has also established an entropic perspective on the state overall bond multiplicity and its covalent and ionic components. The entropic measure of the overall bond “covalency” has been linked to the conditional entropy (average “noise”) in the molecular information system, while the complementary ionicity descriptor has been identified to reflect the mutual information (information “flow”) in this channel. Specific IT tools for detecting effects of chemical bonds, predicting their spatial localization and chemical multiplicity have also been developed. The direct communications between orbitals reflect the “through-space” bonding component, a result of constructive interference between orbitals, while the indirect (cascade) propagations give rise to “through-bridge” bond orders due to orbital intermediates. This approach has increased our insight and understanding of the information origins of the chemical bonding.

In the present analysis we focus on the state continuity relations for the wavefunction modulus and phase components, as well as on the balance equations establishing the net productions of the resultant information and entropy descriptors. The nonclassical origins of the information sources in quantum systems are emphasized and the quantum mixed character of effective states of molecular fragments and interacting reactants is stressed. Reactive complexes involving the donor (basic, B) and acceptor (acidic, A) subsystems are reexamined and a distinction between the interacting and separated (isolated) species are briefly explored. We also investigate the substrate density matrices and inter-reactant electronic communications, which shape the entropic multiplicities of chemical bonds between reactants.

## 2. Probability and Phase Continuities

For simplicity, let us consider a single electron in state |*ψ*(*t*)〉 at time *t*, or the associated (complex) wavefunction in position representation:*ψ*(***r***, *t*) = 〈***r***|*ψ*(*t*)〉 = *R*(***r***, *t*) exp[i*φ*(***r***, *t*)], *φ*(***r***, *t*) ≥ 0.(1)
Its modulus (*R*) and phase (*φ*) components determine the state electron probability and current densities:*p*(***r***, *t*) = *ψ*(***r***, *t*)^*^*ψ*(***r***, *t*) = *R*(***r***, *t*)^2^,(2)
***j****_p_*(***r***, *t*) = [*ħ*/(2*m*i)] [*ψ*(***r***, *t*)^*^ ∇*ψ*(***r***, *t*) − *ψ*(***r***, *t*) ∇*ψ*(***r***, *t*)^*^]  = (*ħ*/*m*) *p*(***r***, *t*) ∇*φ*(***r***, *t*) ≡ *p*(***r***, *t*) ***V***(***r***, *t*). (3)
The effective velocity ***V***(***r***, *t*) of the probability “fluid” measures its current-per-particle and reflects the state phase gradient:***V***(***r***, *t*) = ***j****_p_*(***r***, *t*)/*p*(***r***, *t*) = (*ħ*/*m*) ∇*φ*(***r***, *t*).(4)
In the molecular scenario, an electron is moving in the external potential *v*(***r***) due to the “frozen” nuclear frame of the familiar Born-Oppenheimer approximation. The electronic Hamiltonian
H(***r***) = −[*ħ*^2^/(2*m*)] Δ + *v*(***r***) ≡ T(***r***) + *v*(***r***),(5)
ultimately determines the Schrödinger dynamics of electronic state:i*ħ* (∂*ψ*/∂*t*) = H*ψ*.(6)
This SE of molecular QM also implies specific time evolutions of both components of the complex wavefunction of Equation (1) (see Section 3). The time derivatives of the modulus and phase parts of electronic states ultimately reflect the relevant continuity equations associated with physical descriptors of the particle probability and current densities.

It directly follows from the SE (6) and its complex conjugate that the quantum dynamics implies the probability continuity relation expressing the vanishing source of this distribution:*σ_p_* ≡ *dp*/*dt* = ∂*p*/∂*t* + ∇⋅***j****_p_* = ∂*p*/∂*t* + ***V*** ⋅ ∇*p* = 0  or ∂*p*/∂*t* = 2*R* (∂*R*/∂*t*) = −∇⋅***j****_p_* = −∇*p* ⋅ ***V*** = −(2*R* ∇*R*) ⋅ ***V***. (7)
This equation, thus, expresses the time evolution of the state modulus component:∂*R*/∂*t* = −∇*R* ⋅ ***V***,(8)
while the associated phase dynamics reads:∂*φ*/∂*t* = [*ħ*/(2*m*)] [*R*^−1^Δ*R* − (∇*φ*)^2^] − *v*/*ħ*.(9)
The inflow of probability flux in Equation (7),
∇⋅ ***j****_p_* = ∇*p* ⋅ ***V*** + *p* ∇⋅***V*** = ∇*p* ⋅***V***,(10)
also implies the vanishing divergence of the velocity field ***V***:∇⋅***V*** = (*ħ*/*m*) Δ*φ* = 0  or  Δ*φ* = 0.(11)

In the probability continuity relation of Equation (7),
∂*p*(***r***, *t*)/∂*t* = −∇⋅ ***j****_p_*(***r***, *t*) + *σ_p_*(***r***, *t*)
the negative divergence term, −∇⋅ ***j****_p_*(***r***, *t*), represents the local probability outflow and *σ_p_*(***r***, *t*) stands for its (vanishing) local “source”. The total time derivative of the distribution *p*(***r***, *t*) = *p*[***r***(*t*), *t*], thus, determines the vanishing net production of *p*(***r***, *t*):*σ_p_*(***r***, *t*) = *dp*(***r***, *t*)/*dt* = ∂*p*[***r***(*t*), *t*]/∂*t* + (*d**r***/*dt*) ⋅ ∂*p*(***r***, *t*)/∂***r***         = ∂*p*(***r***, *t*)/∂*t* + ***V***(***r***, *t*) ⋅∇*p*(***r***, *t*) = ∂*p*(***r***, *t*)/∂*t* + ∇⋅ ***j****_p_*(***r***, *t*) = 0.(12)
This total time derivative measures the rate of change in an infinitesimal volume element of probability fluid moving with velocity ***V*** = *d**r***/*dt*, whereas the partial derivative ∂*p*[***r***(*t*), *t*]/∂*t* refers to a volume element around the fixed point in space.

One also realizes that the effective velocity ***V*** of the probability current, ***j****_p_* = *p**V*****,** also determines the phase flux
***j**_φ_* = *φ **V***(13)
and its divergence (see Equation (11)):∇⋅***j**_φ_* = ∇*φ* ⋅***V*** = (*ħ*/*m*) (∇*φ*)^2^.(14)
This complementary flow descriptor ultimately generates a nonvanishing phase source:*σ_φ_* ≡ *dφ*/*dt* = ∂*φ*/∂*t* + ∇⋅ ***j****_φ_* = ∂*φ*/∂*t* + ***V*** ⋅ ∇*φ* ≠ 0.(15)
Using Equation (9) finally gives:*σ_φ_* = [*ħ*/(2*m*)][*R*^−1^Δ*R* + (∇*φ*)^2^] − *v*/*ħ*.(16)
To summarize, the effective velocity of the probability current also determines the phase flux in molecules. The source (net production) of the classical (probability) variable of the electronic state identically vanishes, while that of its nonclassical (phase) component remains finite. The phase source is seen to be determined by both wavefunction components and the external potential due to the system nuclei.

## 3. Resultant Entropy/Information Descriptors and State Continuity

Equation (1) identifies the following (additive) components of the wavefunction logarithm:ln*ψ* = ln*R* + i*φ*.(17)
Together they generate the so-called resultant measures of the global and gradient contents of the state overall entropy or information [53,54,55,56,57,58,59,60,61,62]. For example, for the given time *t*, the complex global entropy descriptor [59,62], the quantum expectation value of the (non-Hermitian) multiplicative operator of the complex global entropy, *S*(***r***) = −2ln*ψ*(***r***),
     *S*[*ψ*] = 〈*ψ*|*S*|*ψ*〉 = −2〈*ψ*|ln*ψ*|*ψ*〉= −2∫*p*(***r***) ln*ψ*(***r***) *d**r*****≡** ∫*p*(***r***) *S*(***r***) *d**r*** = −∫*p*(***r***) [ln*p*(***r***) + 2i*φ*(***r***)] *d**r*** ≡ *S*[*p*] + i *S*[*φ*], (18)
where *S*(***r***) stands for the entropy density per electron, combines the classical (Shannon) entropy *S*[*p*] **≡** ∫*p*(***r***)*S_p_*(***r***)*d**r*** as its real part, and the nonclassical phase supplement *S*[*φ*] **≡** ∫*p*(***r***)*S_φ_*(***r***)*d**r***, which determines its imaginary component. The latter reflects the state average phase *φ*[*ψ*] = ∫*p*(***r***)*φ*(***r***)*d**r***: *S*[*φ*] = −2*φ*[*ψ*].

The corresponding Fisher-type gradient measure of the state resultant information *I*[*ψ*] is defined by quantum expectation *I*[*ψ*] = 〈*ψ*|I|*ψ*〉 of the (Hermitian) operator in position representation,
I(***r***) = −4Δ = (8*m*/*ħ*^2^) T(***r***) ≡ σ T(***r***), (19)
   *I*[*ψ*] = 4∫*p*(***r***){[∇ln*R*(***r***)]^2^ − [i∇*φ*(***r***)]^2^} *d**r*****≡** ∫*p*(***r***) *I*(***r***) *d**r***     = 4∫{[∇*R*(***r***)]^2^ + [*R*(***r***) ∇*φ*(***r***)]^2^} *d**r*** ≡ *I*[*R*] + *I*[*φ*]= ∫*p*(***r***) {[∇ln*p*(***r***)]^2^ + 4[∇*φ*(***r***)]^2^} *d**r***        = ∫*p*(***r***)^−1^[∇*p*(***r***)]^2^*d**r*** + 4∫*p*(***r***)[∇*φ*(***r***)]^2^*d**r*** ≡ *I*[*p*] + *I*[*φ*].(20)

The associated gradient measure of resultant entropy
*M*[*ψ*] = 〈*ψ*|M|*ψ*〉 **≡** ∫*p*(***r***) *M*(***r***) *d**r***
reads:*M*[*ψ*] = 4∫*p*(***r***){[∇ln*R*(***r***)]^2^ + [i∇*φ*(***r***)]^2^} *d**r*** = *I*[*p*] − *I*[*φ*]   = 4∫{[∇*R*(***r***)]^2^ − [∇*φ*(***r***)]^2^} *d**r*** ≡ *M*[*R*] + *M*[*φ*]     = ∫*p*(***r***){[∇ln*p*(***r***)]^2^ − 4[∇*φ*(***r***)]^2^} *d**r*** ≡ *M*[*p*] + *M*[*φ*].(21)
The generalized information measure *I*[*ψ*], related to the average kinetic energy *T*[*ψ*] = 〈*ψ*|T|*ψ*〉, combines the classical Fisher information in probability distribution, *I*[*p*] **≡** ∫*p*(***r***)*I_p_*(***r***)*d**r***, and its nonclassical complement *I*[*φ*] **≡** ∫*p*(***r***)*I_φ_*(***r***)*d**r***, due to inhomogeneities in the state phase distribution.

These equations confirm a symmetrical role played by the additive components of Equation (17) in generating the overall entropy and information content in the quantum electronic state. The resultant distribution of the local resultant gradient information reflects the density of electronic kinetic energy. One also observes that densities-per-electron of the gradient information and complex global entropy are mutually related:*I*(***r***) = |∇*S*(***r***)|^2^.
Thus, the gradient of the latter constitutes the (coherent) quantum amplitude of the former. Both resultant densities of an entropic content of the electronic state are seen to include the nonclassical (phase/velocity) complements of the classical (modulus/probability) contributions. The preceding relation constitutes a natural (complex) generalization of the corresponding classical link between local information and entropy descriptors of Fisher and Shannon:*I_p_*(***r***) = [∇*S_p_*(***r***)]^2^.
It should also be observed that a (noncoherent) classical density of Shannon’s global entropy,
*S_p_*(***r***) = − 2 ln*R*(***r***) = − ln*p*(***r***) = Re *S*(***r***),
is devoid of any phase content.

A reference to Equation (19) indicates that the state information functional *I*[*ψ*] is proportional to the average kinetic energy of electrons *T*[*ψ*] = 〈*ψ*|T|*ψ*〉,
*I*[*ψ*] = (8*m*/*ħ^2^*) *T*[*ψ*] = *σT*[*ψ*],
determined by the quantum operator of Equation (5):*T*(***r***) = − [*ħ*^2^/(2*m*)] Δ = [*ħ*^2^/(8*m*)] *I*(***r***) ≡ *σ*^−1^*I*(***r***).
One also recalls that the average electronic energy in state |*ψ*(*t*)〉 combines the following components:*E*[*ψ*] = 〈*ψ*|H|*ψ*〉 = −[*ħ*^2^/(2*m*)] ∫[*R*Δ*R* − *R*^2^(∇*φ*)^2^] + ∫*R*^2^*v d**r***       = [*ħ*^2^/(2*m*)] ∫[(∇*R*)^2^ + *R*^2^(∇*φ*)^2^] *d**r*** + ∫*R*^2^*v d**r*****≡***T*[*ψ*] + *V_ne_*[*ψ*],
where we have used the relevant integration by parts and *V_ne_*[*ψ*] denotes the average energy of the electron nuclei attraction.

Expressing SE in terms of the state modulus and phase components *R* and *φ* gives:i (∂ln*R*/∂*t*) − (∂*φ*/∂*t*) = −[*ħ*/(2*m*)] [*R*^−1^Δ*R* + 2i∇ln*R* ⋅∇*φ* − (∇*φ*)^2^] + *v*/*ħ*.(22)
Its imaginary parts determine the continuity equation for the wavefunction modulus component (see Equation (8)),
∂ln*R*/∂*t* = −***V***⋅∇ln*R* = −∇⋅[(ln*R*)***V***] ≡ −∇⋅***j****_R_*,(23)
where we have introduced the modulus flux
***j****_R_* = (ln*R*) ***V***(24)
associated with real part of the state logarithm (17). The real components of Equation (22) similarly recover the phase dynamics of Equation (9):∂*φ*/∂*t* = − ∇⋅ ***j****_φ_* + *σ_φ_* = [*ħ*/(2*m*)] [*R*^−1^Δ*R* − (∇*φ*)^2^] − *v*/*ħ*.(25)
One also observes that combining the preceding equation (multiplied by i) with the modulus continuity of Equation (23) gives the state logarithmic continuity relation
∂ln*ψ*/∂*t* = ∂ln*R*/∂*t* + i ∂*φ*/∂*t* = −∇⋅(***j****_R_* + i***j****_φ_*) + i*σ_φ_*    = −∇⋅ (***V*** ln*ψ*) + *σ*_ln*ψ*_ ≡ −∇⋅***J***_ln*ψ*_ + σ_ln*ψ*_.(26)
This equation can be also recast to express the state logarithmic source:*σ*_ln*ψ*_ ≡ *d*ln*ψ*/*dt* = *ψ*^−1^ (*dψ*/*dt*) = ∂ln*ψ*/∂*t* + ***V***⋅∇ln*ψ*         = ∂ln*ψ*/∂*t* + ∇⋅(***V*** ln*ψ*) = ∂ln*ψ*/∂*t* + ∇⋅ ***J***_ln*ψ*_ = *σ*_ln*ψ*_ = i*σ_φ_*.(27)
The (complex) logarithmic continuity relation further emphasizes a classical (real) character of the modulus and probability descriptors and a nonclassical (imaginary) nature of the phase and current state variables. It introduces the (complex) wavefunction current
***J***_ln*ψ*_ = ***j****_R_* + i***j****_φ_* = ***V*** ln*ψ*(28)
and identifies the (nonclassical) state source,
*σ*_ln*ψ*_ = i*σ_φ_*,(29)
stressing the phase origin of the state production of its overall information and entropy density. This combined treatment also reveals two independent sources of the resultant entropy and information descriptors of Equations (18), (20), and (21), the additive components of the logarithmic separation of Equation (17).

## 4. Integral Productions of Information and Entropy Descriptors

It is also of interest to examine the integral sources of the resultant measures of the quantum global entropy or gradient information. For the average production of the state complex entropy,
*S*[*ψ*] = −2∫*p* ln*ψ d**r***
one finds:*σ_S_*[*ψ*] = *dS*[*ψ*]/*dt* = −2 ∫{(*dp*/*dt*) ln*ψ* + *p* (*d*ln*ψ*/*dt*)} *d**r*** = −2 ∫*p σ*_ln*ψ*_*d**r***,(30)
where we have recognized the probability continuity of Equation (7). Using Equation (30), then, finally gives
*σ_S_*[*ψ*] = −2∫*pσ*_ln*ψ*_*d**r*** = −2i ∫*pσ_φ_ d**r*** ≡ −2i*σ_φ_*[*ψ*].(31)
This expression again confirms the nonclassical (phase) origin of the complex entropy source, which reflects the average of the local production *σ_φ_*[*ψ*] of the state phase (see Equation (16)).

The integral production *σ_S_*[*ψ*] of state overall entropy
*S*[*ψ*] ≡ ∫*p*(***r***) *S*(***r***) *d**r***
can be also discussed in terms of the local source contribution per electron, *σ_S_*(***r***),
*σ_S_*[*ψ*] ≡ ∫*σ_p_*(***r***) *S*(***r***) *d**r*** + ∫*p*(***r***) *σ_S_*(***r***) *d**r*** = ∫*p*(***r***) *σ_S_*(***r***) *d**r***,
and the associated continuity relation
*σ_S_*(***r***) = ∂*S*(***r***)/∂*t* + ∇⋅***J****_S_*(***r***),  ***J****_S_*(***r***) = *S*(***r***) ***V***(***r***) = [*S*(***r***)/*p*(***r***)] ***j****_p_*(***r***),
where ***J****_S_*(***r***) stands for the density of entropy current carried by the probability flux ***j****_p_*(***r***). Then, it again follows from Equations (16) and (18) that the local source of the complex entropy has purely nonclassical, phase origin: *σ_S_*(***r***) = −2i*σ_φ_*(***r***).

One similarly determines the corresponding total time derivatives of the overall gradient measures of the state resultant information (Equation (20)) and entropy (Equation (21)):  *I*[*ψ*] = ∫*p*[(∇ln*p*)^2^ + 4(∇*φ*)^2^] *d**r*** = *I*[*p*] + *I*[*φ*]  and *M*[*ψ*] = ∫*p*[(∇ln*p*)^2^ − 4(∇*φ*)^2^] *d**r*** = *M*[*p*] + *M*[*φ*].(32)
Using the probability and phase continuity relations, one ultimately obtains the following expressions for the average productions of these state functionals:    *σ_I_*[*ψ*] = *dI*[*ψ*]/*dt* = ∫(*dp*/*dt*) (*δI*[*p*]/*δp*) *d**r*** + *dI*[*φ*]/*dt* = *dI*[*φ*]/*dt*   = 4∫{(*dp*/*dt*)(∇*φ*)^2^ + 2*p* ∇*φ* ⋅ *d*(∇*φ*)/*dt*} *d****r***   = 8∫*p*∇*φ* ⋅∇(*dφ*/*dt*) *d**r*** = 8∫*p*∇*φ* ⋅∇*σ_φ_ d**r***≡ −*σ_M_*[*ψ*] = −*dM*[*ψ*]/*dt* = −*dM*[*φ*]/*dt*.(33)

These expressions reveal the complementary character of the gradient information and entropy, with the positive source of one implying the negative production of another. The time derivatives of these overall functionals manifest the nonclassical (phase) origins of the molecular productions of electronic gradient entropy and information descriptors [64]. These integral sources can be also expressed in terms of the electron current density ***j****_p_* of Equation (3):*σ_I_*[*ψ*] = (8*m*/*ħ*) ∫***j****_p_* ⋅∇*σ_φ_ d**r*** = −*σ_M_*[*ψ*].(34)
In close analogy to irreversible thermodynamics [83], they are seen to be determined by the product of local flux and affinity densities, ***j****_p_*(***r***) and ∇*σ_φ_*(***r***), respectively. Using Equations (11) and (16), finally gives the following explicit expression for the phase-source gradient: ∇*σ_φ_* = [*ħ*/(2*m*)]{*R*^−2^[*R*∇^3^*R* − (∇*R*) Δ*R*] + 2(∇*φ*) Δ*φ*} − ∇*v*/*ħ*= [*ħ*/(2*m*)][*R*^−1^ ∇^3^*R* − (∇ln*R*) *R*^−1^Δ*R*] − ∇*v*/*ħ*.(35)
Thus, it follows that only the wavefunction modulus and shape of the external potential influence the affinity factor in the resultant information and entropy production of Equation (37). It should be recalled, however, that the phase gradient, ∇*φ*, determines the flux factor, ***j****_p_*, in this product.

The integral source of the gradient information (Equation (36)) can be also interpreted in terms of the local information source *σ_I_*(***r***):*σ_I_*[*ψ*] = *dI*[*ψ*]/*dt* = 8∫*p*(***r***)∇*φ*(***r***) ⋅∇*σ_φ_*(***r***) *d**r*** ≡ ∫*p*(***r***) *σ_I_*(***r***) *d**r***.(36)
The local continuity equation for the information density *I*(***r***) of Equation (20),
*σ_I_*(***r***) ≡ *dI*(***r***)/*dt* = ∂*I*(***r***)/∂*t* + ∇⋅***J****_I_*(***r***) = 8∇*φ*(***r***) ⋅∇*σ_φ_*(***r***)  or  ∂*I*(***r***)/∂*t* = −∇⋅***J****_I_*(***r***) + *σ_I_*(***r***) = [8∇*σ_φ_*(***r***) − (*ħ*/*m*) *I*(***r***)] ⋅ ∇*ϕ*(***r***),(37)
also involves the information current density:***J****_I_*(***r***) = *I*(***r***) ***V***(***r***) = (*ħ*/*m*) *I*(***r***) ∇*ϕ*(***r***),  ∇⋅***J****_I_*(***r***) = (*ħ*/*m*) ∇*I*(***r***) ⋅ ∇*ϕ*(***r***).(38)
The local continuity relations of Equation (37) again emphasize the nonclassical (phase) origin of the information source. One observes that only a presence of the state local phase contribution generates a finite probability flow and nonvanishing information production.

## 5. Isolated/Interacting and Open/Closed Subsystems

Consider the simplest case of the two-electron reactive complex consisting of the B^+^ (base, electron donor) and A^+^ (acid, electron acceptor) subsystems, each containing a single electron, *N*_A_^0^ = *N*_B_^0^ = 1, at the polarization (P) stage of the reactive system [63,64,65,67,68,69,70,71]:R^+^(1, 2) = [A^+^(1) | B^+^(2)],  *N* = *N*_A_^0^ + *N*_B_^0^ = 2.(39)
This model system involves the mutually-closed substrates, at a finite distance R_AB_ between the two subsystems with both (interacting) electrons, *g*(1,2) = *r*_1,2_^−1^(a.u.), moving in the external potential *v* = *v*_A_ + *v*_B_, due to the fixed nuclei of both geometrically “frozen” reactants. Their infinite separation, R_AB_→∞, results in the sum of isolated reactants {X^0^},
R^∞^ = A^0^(1) + B^0^(2) ≡ R^0^,
while the mutual opening of these molecular fragments in R* = (A*¦B*), at a finite distance R_AB_, results in the global equilibrium state of R as a whole, after the optimum B→A charge transfer (CT):R*(1, 2) = A*(1, 2) + B*(1, 2).
As indicated above, such open interacting subsystems assume an effective two-electron character, since in R* the two electrons are indistinguishable. Therefore, both mutually open subsystems effectively explore the probability distribution *p*(***r***) of the whole complex. The equilibrium subsystems {X*} are, then, characterized by the subsystem densities {*ρ*_X_*(***r***)} exhibiting fractional average numbers of electrons {*N*_X_* = ∫*ρ*_X_*(***r***)*d**r***}. They generate the final, equilibrium molecular distribution of the whole reactive complex,
*ρ*_A_*(***r***) + *ρ*_B_*(***r***) = *ρ*_AB_(***r***) ≡ *ρ*(***r***) = *N p*(***r***),(40)
and define the optimum amount of the B→A CT:*N*_CT_ = *N*_A_* − *N*_A_^0^ = *N*_B_^0^ − *N*_B_ *,{*N*_X_^0^ = 1, *N*_X_* = ∫*ρ*_X_*(***r***)d***r***(fractional)}.(41)
One further recalls that in theory of chemical reactivity this (global) *N*_CT_ measure results from the electronegativity-equalization (EE) considerations [84,85,86,87,88,89,90], based upon the chemical potential [91,92,93,94,95] and hardness/softness [96] or Fukui function [97] derivative descriptors of the (mutually-closed) polarized subsystems in R^+^.

This model scenario, thus, involves the one-electron Hamiltonians {h^0^(X)} of the isolated, infinitely separated fragments {X^0^(*i*)} in R^∞^ (see Equation (5)),
   h^0^(A) = − [*ħ*^2^/(2*m*)] Δ_1_ + *v*_A_(1) ≡ T(1) + *v*_A_(1) = h_A_^0^(1)  andh^0^(B) = − [*ħ*^2^/(2*m*)] Δ_2_ + *v*_B_(2) ≡ T(2) + *v*_B_(2) = h_B_^0^(2).(42)
Their eigenvalue problems,
h^0^(X) *ψ_w_*^X^ = *ε_w_*^0^*ψ_w_*^X^,  ∫*ψ*_*w*′_^X*^(*i*) *ψ_w_*^X^(*i*) *d**r**_i_* ≡ 〈*w*′(X)|*w*(X)〉*_i_* = *δ_w_*_,*w*′_,(43)
define the (one-electron) othonormal bases of the alternative complete sets of stationary states in isolated fragments, also capable of representing any (two-electron) state Ψ(A, B) = Ψ(1, 2) of the whole reactive complex. For example, in the A expansion:Ψ(1, 2) = ∑*_w_ χ_w_*^A^(2) *ψ_w_*^A^(1),(44)
*χ_w_*^A^(2) = ∫*ψ_w_*^A^(1)^*^ Ψ(1, 2) *d**r***_1_ ≡ 〈*w*^A^|Ψ〉_1_,〈Ψ|Ψ〉_1,2_ = ∑*_w_* ∑**_*w*′_ 〈*χ_w_*^A^|*χ*_*w*′_^A^〉_2_ 〈*w*′(A)|*w*(A)〉_1_ = ∑*_w_* 〈*χ_w_*^A^|*χ_w_*^A^〉_2_ = 1.

The mutually- and externally-open, interacting parts of this model reactive system are in the mixed states described by the corresponding density matrices of subsystems [82]. Indeed, for the mutually-open, interacting fragments, a simple product representation of this (pure) quantum state,
Ψ(1, 2) = *ψ*_A_(1) *ψ*_B_(2),(45)
with each reactant described by the substrate wavefunction that is dependent exclusively on its own internal coordinates, is not available. It exists only for the (disentangled) states of noninteracting subsystems in R^∞^ ≡ R^0^,
Ψ^0^(1, 2) = *ψ*_A_^0^(1) *ψ*_B_^0^(2), 
and the distinguishable electrons attributed to the mutually-closed (*c*) reactants in the polarized reactive system R^+^ = (A^+^|B^+^) ≡ R*_c_*,
Ψ*_c_*^+^(1, 2) = *ψ*_A_^+^(1) *ψ*_B_^+^(2).
This product wavefunction is replaced by the corresponding Slater determinant |*ψ*_A_^+^
*ψ*_B_^+^|, when the two polarized subsystems become mutually open (*o*) in R* = (A*¦B*) ≡ R*_o_* thus making the two electrons indistinguishable:Ψ*_o_*^+^(1, 2) ≡ |*ψ*_A_^+^*ψ*_B_^+^| = 2^−1/2^ [*ψ*_A_^+^(1) *ψ*_B_^+^(2) − *ψ*_A_^+^(2) *ψ*_B_^+^(1)].

The (two-electron) Hamiltonians, describing the interacting subsystems in R*_c_*^+^ at finite distances between reactants, read:   H_A_^+^(1, 2) = T(1) + *v*(1) + *g*(1, 2) = h_A_^0^(1) + [*v*_B_(1) + *g*(1, 2)]  and H_B_^+^(1, 2) = T(2) + *v*(2) + *g*(1, 2) = h_B_^0^(2) + [*v*_A_(2) + *g*(1, 2)].(46)
The complete sets of their stationary states {ΘuX(1, 2)},
H_X_^+^(1, 2) Θ*_u_*^X^(1, 2) = *e_u_*^X^ Θ*_u_*^X^(1, 2),  〈Θ*_u_*^X^|Θ*_u_*_′_^X^〉_1,2_ = *δ_u_*_,*u*′_,(47)
then, define the alternative (two-electron) bases for expanding the “molecular” state Ψ(1, 2):Ψ(1, 2) = ∑*_u_ C_u_*^X^ Θ*_u_*^X^(1, 2),  *C_u_*^X^ = ∫∫Θ*_u_*^X^(1, 2)^*^ Ψ(1, 2) *d**r***_1_*d**r***_2_ ≡ 〈Θ*_u_*^X^|Ψ〉_1_,_2_.(48)
The equilibrium reactive complex, R* = (A*¦B*), at a finite distance between the two mutually-open substrates, corresponds to the electronic Hamiltonian of reactive complex as a whole,
          H(1, 2) = [T(1) + T(2)] + [*v*(1) + *v*(2)] + *g*(1, 2) ≡ T(1, 2) + *V*(1, 2) + *g*(1, 2)                = [H_A_^0^(1) + H_B_^0^(2)] + [*v*_B_(1) + *v*_A_(2) + *g*(1, 2)] ≡ H_R_^0^(1, 2) + *h*(1, 2)  = H_A_^+^(1, 2) + H_B_^+^(1, 2) − *g*(1, 2), (49)
where *h*(1, 2) stands for the overall perturbation relative to the reference Hamiltonian H_R_^0^(1, 2) in the separated reactant limit (SRL). In Appendix A the energetic implications of the mutual opening of reactants are briefly examined using the 1st—order perturbation theory.

This molecular Hamiltonian determines the stationary states of the whole reactive complex R*:H(1, 2) Ψ*_s_*(1, 2) = *E_s_* Ψ*_s_*(1, 2).(50)
Their phase component is purely time dependent,
Ψ*_s_*(1, 2) = *R_s_*(1, 2) exp[i*Φ**_s_*(t)],*Φ**_s_*(t) = − (*E_s_*/*ħ*) *t* = − *ω_s_ t*,(51)
thus, giving rise to the vanishing phase gradient, and hence zero probability current.

A general “molecular” state Ψ(***r***_1_, ***r***_2_) ≡ Ψ(1, 2) of two indistinguishable electrons, which determines the electron density
*ρ*(***r***) = 2∫|Ψ(***r***,***r***_2_)|^2^*d**r***_2_ = 2*p*(***r***),(52)
also characterizes all equilibrium reactants {X*} in R* = (A*¦B*), since all mutually-open fragments explore the same “molecular” probability distribution:*p*_A_*(***r***) = *ρ*_A_*(***r***)/*N*_A_* = *p*_B_*(***r***) = *ρ*_B_*(***r***)/*N*_B_* = *p*(***r***) = *ρ*(***r***)/*N*.(53)
Their (mixed) quantum states are represented by the corresponding density operators, for example, those corresponding to the applied (external) thermodynamic conditions. The reactive system coupled to an external heat bath *B*(*T*) and electron reservoir *R*(*μ*) would be represented by the equilibrium *grand*-ensemble establishing the statistical mixture of {Ψ*_s_*(1, 2)}. The state probabilities are then related to the absolute temperature *T* of *B* and the chemical potential *μ* of *R* [94,95].

Expanding a general (pure) state Ψ(1, 2) of R* in the stationary molecular basis {Ψ*_s_*(1, 2)} gives:Ψ(1, 2) = ∑*_s_ D_s_* Ψ*_s_*(1, 2),  *D_s_* = 〈Ψ*_s_*|Ψ〉_1,2_.(54)
In this molecular state the expectation value of a property *F*_A_ of subsystem A, represented by the associated quantum operator F_A_(1), is given by an ensemble-average expression including the partial (fragment) trace operation and the subsystem density matrix **ρ**(A) [82,98]. Indeed, using the fragment expansion of Equation (44) gives
〈*F*_A_〉_Ψ_ = 〈Ψ|F_A_|Ψ〉 = ∑*_w_* ∑**_*w*′_ [∫*χ*_*w*′_^A^(2)^*^*χ_w_*^A^(2) *d**r***_2_] [∫*ψ*_*w*′_^A^(1)* F_A_(1) *ψ_w_*^A^(1) *d**r***_1_] = ∑_*w*_ ∑_*w*′_〈*χ*_*w*′_^A^|*χ_w_*^A^〉_2_ 〈*w*′(A)|F_A_|*w*(A)〉_1_   ≡ ∑*_w_* ∑**_*w*′_*ρ_w_*,**_*w*′_(A) *F_w_*_′,*w*_(A) ≡ tr_A_ [**ρ**(A) **F**(A)].(55)
The diagonal element of the above subsystem density matrix (see the normalization condition of Equation (44)),
*ρ_w_*,_*w*_(A) = ∫|*χ_w_*^A^(2)|^2^*d**r***_2_ = *P*[*ψ_w_*^A^|Ψ],(56)
∑*_w_ P*[*ψ_w_*^A^|Ψ] = ∑*_w_* 〈*χ_w_*^A^|*χ_w_*^A^〉_2_ = 1
measures conditional probability *P*[*ψ_w_*^A^|Ψ] of observing in Ψ the subsystem state *ψ_w_*^A^ = *R_w_*^A^exp(i*φ_w_*^A^). These probabilities define the effective density operator of A in the molecular state Ψ,
d^A^ = ∑*_w_* |*ψ_w_*^A^〉 *P*[*ψ_w_*^A^|Ψ] 〈*ψ_w_*^A^|,(57)
which determines the effective (mixed) state of this fragment in the reactive system. Its representation in the basis {*ψ_w_*^A^} of Equation (43) is diagonal:**d**^A^ = {*d_w_*,**_*w*′_^A^ = *P*[*ψ_w_*^A^|Ψ] *δ_w_*_,*w*′_}.(58)
Therefore, by selecting in Equation (55) F_A_ = *φ*_A_, one obtains the following expression for the representative average phase of A in Ψ:〈*φ*_A_〉_Ψ_ = 〈Ψ|*φ*_A_|Ψ〉 = ∑*_w_ P*[*ψ_w_*^A^|Ψ] *φ_w_*^A^.(59)

## 6. Equidensity Orbital Systems

As an illustration, consider the EO configurations defined by Slater determinants of orbitals conserving the specified molecular probability distribution. In HZM construction [76,77] of modern DFT [40,41,42,43,44,45], of the wavefunctions yielding the prescribed electron density, one introduces the plane-wave type EO,
          *ϕ_l_*[*p*; ***r***] = [*ρ*(***r***)/*N*]^1/2^ exp{i[***q****_l_*⋅***f***(***r***) + *φ*(***r***)]} ≡ *p*(***r***)^1/2^ exp{i[*F_l_*(***r***) + *φ*(***r***)]} ≡ *p*(***r***)^1/2^ exp[i*Φ*_*l*_(***r***)] ≡ φ_*l*_(***r***),(60)
which exactly reproduce the system probability distribution *p*(***r***). The density-dependent vector function ***f***(***r***) = *f_x_*(***r***) ***i*** + *f_y_*(***r***) ***j*** + *f_z_*(***r***) ***k*** =***f***[*p*;***r***], for which the Jacobian determinant
(61)$$\frac{\partial f}{\partial r}=\left|\begin{array}{ccc} \frac{\partial f_{x}}{\partial x} & 0 & 0\\ \frac{\partial f_{x}}{\partial y} & \frac{\partial f_{y}}{\partial y} & 0\\ \frac{\partial f_{x}}{\partial z} & \frac{\partial f_{y}}{\partial z} & \frac{\partial f_{z}}{\partial z} \end{array}\right|=\left(\frac{\partial f_{x}}{\partial x}\right)\left(\frac{\partial f_{y}}{\partial y}\right)\left(\frac{\partial f_{z}}{\partial z}\right)={(2\pi )}^{3}p(r)$$
then assures the orbital orthonormality:(62)$$\begin{array}{ll} \int_{-\infty }^{\infty }\phi_{q^{\prime }}^{*}(r)\phi_{q}^{}(r)dr & =\int_{-\infty }^{\infty }e^{i(q-q^{\prime })\cdot f(r)}p(r)dr\\ & =\frac{1}{{(2\pi )}^{3}}\int_{-\infty }^{\infty }e^{i(q-q^{\prime })\cdot f(r)}\frac{\partial f}{\partial r}dr=\frac{1}{{(2\pi )}^{3}}\int_{-\infty }^{\infty }e^{i(q-q^{\prime })\cdot f}df=\delta (q-q^{\prime }). \end{array}$$
Here, ***q****_l_* = (*q_l_*_,*x*_
***i*** + *q_l_*_,*y*_
***j*** + *q_l_*_,*z*_
***k***) denotes the (constant) reduced momentum (wave number) vector of EO and *Φ**_l_*(***r***) stands for its resultant phase. The latter is defined by the sum of orthogonality phase *F**_q_***(***r***) and its local “thermodynamic” supplement *φ*(***r***), common in all occupied EO of the system electron configuration under consideration:Ψ[*p*; *N*] = |*φ*_1_[*p*] *φ*_2_[*p*] … *φ_N_*[*p*]|,  ***q***_1_[*p*] ≠ ***q***_2_[*p*] ≠ … ≠ ***q****_N_*[*p*].(63)

Notice, that in this HZM representation all orbital components are described by the local resultant phases {*Φ**_l_*(***r***) = *Φ**_l_*[*p*; ***r***], *l* = 1, 2, …, *N*} originating from the same overall probability density *p*(***r***). The resultant local phase *Φ**_l_*(***r***) also generates the associated orbital current of Equation (3):***j****_l_*(***r***) = (*ħ*/*m*) *p*(***r***) ∇*Φ**_l_*[*p*; ***r***].(64)
The optimum “thermodynamic” contribution *φ^opt.^*(***r***), common to all occupied EO reconstructing the given electron density *ρ*(***r***) = *N p*(***r***), is determined from the subsidiary minimum information principle [53,54,55,56,57]. It relates this phase contribution to the average wave vector in the configuration under consideration [78,79]:*φ^opt.^*(***r***) = *φ*[*p*; ***r***] = − 〈***q***[*p*]〉 ⋅***f***[*p*; ***r***],  〈***q***[*p*]〉 = *N*^−1^ ∑*_l_****q****_l_*[*p*],(65)
where the summation extends over all occupied EO. Then, the resultant EO phases are shaped by displacements {*δ**q**_l_* =***q****_l_*[*p*] − 〈***q***[*p*]〉} of the the orbital wave vectors {***q****_l_*[*p*]} from the configuration average vector of the preceding equation:{*Φ**_l_^opt.^*[*p*; ***r***] = *δ**q**_l_* ⋅***f***[*p*; ***r***]}.(66)
These resultant phases, then, give rise to the vanishing overall current ***j****_p_*(***r***) in electron configuration (63), the sum of equilibrium EO currents
***j****_l_^opt.^*(***r***) = (*ħ*/*m*) *p*(***r***) *δ**q**_l_* ⋅∇***f***[*p*; ***r***],  *l* = 1, 2, …, *N*,(67)
***j****_p_^opt.^*(***r***) = ∑*_l_****j****_l_^opt.^*(***r***) = (*ħ*/*m*) *p*(***r***) (∑*_l_ δ**q**_l_*) ⋅∇***f***[*p*; ***r***] = 0,(68)
since
∑*_l_ δ**q**_l_* = ∑*_l_**q**_l_* − *N*〈***q***[*p*]〉 = 0.(69)

The occupied EO in the HZM product state
Ψ^+^(*N*) = *φ*_1_^+^(1) *φ*_2_^+^(2) … *φ*_N_^+^(*N*) ≡ [*φ*_1_^+^(1) | *φ*_2_^+^(2) | … |*φ_N_*^+^(*N*)] ≡ Ψ*_c_*(*N*)(70)
represent the *closed* (*c*) orbital system Ψ*_c_*(*N*), with each EO containing a single (distinguishable) electron, {*n_l_*^+^ = l}. These (disentangled) nonbonded orbital components are distinguished by their EO phases, with different wave vectors attributed to each orbital (see Equation (63)).

One can also envisage its *open* (*o*) (bonded, entangled) analog of this *N*-orbital system, described by the Slater determinant of Equation (63):Ψ*(*N*) = |*φ*_1_* *φ*_2_* … *φ_N_**| = (φ_1_*¦φ_2_*¦ … ¦φ_N_*) ≡ Ψ*_o_*(*N*).(71)

This mutual opening of EO in Ψ*(*N*), although still precluding the net electron flows between orbitals, due to the limiting occupations {*n_l_* * = 1}, now formally opens electronic *exchanges*, since in the determinantal state all electrons are indistinguishable (see also Appendix A).

The mutually-open state of orbital subsystems can also involve the external (thermodynamic) coupling of these orbital components to the heat bath *B*(*T*) and (“molecular”) electron reservoir *R*(*μ*) in the (macroscopic) composite system:M(*N*) = [*B*(*T*) ¦ *φ*_1_*(*μ*, *T*) ¦ *φ*_2_*(*μ*, *T*) ¦ … ¦*φ_N_**(*μ*, *T*)¦*R*(*μ*)].(72)
This mutual and external opening of EO fragments in M(*N*) implies their effectively “bonded” (entangled) character. It is reflected by their fractional orbital occupations {0 < *n_l_**(*μ*, *T*) < 1} marking partial electron outflows to the initially unoccupied (virtual) EO. Indeed, the externally open, thermodynamic-orbital fragments must be described by the statistical mixture of EO states {|*φ_l_*〉}, defined by the equilibrium density operator
D(*μ*, *T*) = ∑*_l_* |*φ_l_*〉 *P_l_*(*μ*, *T*) 〈*ϕ_l_*|,  ∑*_l_ P_l_*(*μ*, *T*) = 1,(73)
with the equilibrium orbital probabilities {*P_l_*(*μ*, *T*)} reflecting the applied thermodynamic conditions, i.e., the chemical potential *μ* of the reservoir and the absolute temperature *T* of the heat bath. This mixed state of the mutually-open (bonded, entangled) orbital components corresponds to an equalized (average) phase intensity and a common level of the chemical potential, fixed by the electron reservoir. The equilibrium probability of the *φ_l_* “subsystem” in such an EO *grand*-ensemble is determined by thermodynamic parameters *μ* and *T*, the equilibrium (fractional) orbital occupations {0 < *n_l_** < 1} and orbital energies {*e_l_* = 〈*ϕ_l_*|H|*ϕ_l_*〉}:*P_l_*(*μ*, *T*) = Ξ^EO^(*μ*, *T*)^−1^ exp[*β*(*μn_l_** − *e_l_*)].(74)
Here, Ξ^EO^(*μ*, *T*) = ∑*_l_* exp[*β*(*μn_l_** − *e_l_*)] denotes the EO grand-partition function, *β* = (*k*_B_*T*)^−1^, and *k*_B_ is the Boltzmann constant.

## 7. Electron Communications

Let us now examine the molecular electron communications between states of isolated subsystems in the donor-acceptor reactive system, R = A----B. For specificity and simplicity reasons we again refer to the two-electron scenario of Section 5. The whole information system of the probability scattering between stationary states of isolated reactants in such a molecular complex involves four blocks of conditional probabilities,
**P**_R_ = {**P**(X→Y), X, Y ∈ (A, B)},
defining the internal (diagonal) and external (off-diagonal) blocks of electronic communications within and between reactants, respectively. In Communication Theory of the Chemical Bond [9,10,11,22,23,24,25,26,27,28,29,30,31,32], the former determines the *intra*-reactant bonds, i.e., determine the substrate polarization and activation accompanying the chemical reaction, while the latter reflect the *inter*-reactant bond pattern which directly probes the reactivity behavior. In what follows, we shall focus on this external part of electronic communications alone. For specificity, we, thus, examine the
{*ψ_w_*^A^→*ψ*_*w*′_^B^} or {|*w*(A)〉 → |*w*′(B)〉}
communications described by the molecular probabilities **P**(A→B).

In accordance with SP of QM [99,100], the conditional probability *P*[*w*′(B)|*w*(A)] ≡ *P_w_*_→*w*′_(A→B) of observing the output state |*ψ*_*w*′_^B^〉 ≡ |*w*′(B)〉 of the the “receiver” part B of this partial reactive network, given the input state |*ψ_w_*^A^)〉 ≡ |*w*(A)〉 in its “source” part A, is determined by the squared modulus of the corresponding scattering amplitude *A*[*w*′(B)|*w*(A)] ≡ *A_w_*_→*w*′_(A→B) measuring their mutual projection in the molecular Hilbert space [62,100]:*P_w_*_→*w*′_(A→B) = |*A_w_*_→*w*′_(A→B)|^2^,  *A_w_*_→*w*′_(A→B) = 〈*w*(A)|*w*′(B)〉.(75)
These probabilities satisfy the relevant normalization involving summation over the complete set of all monitoring states in this inter-reactant communication “device”:∑**_*w*′_*P_w_*_→*w*′_(A→B) = ∑**_*w*′_ 〈*w*(A)|*w*′(B)〉 〈*w*′(B)|*w*(A)〉                = 〈*w*(A)|∑**_*w*′_ P**_*w*′_(B)|*w*(A)〉 = 〈*w*(A)|*w*(A)〉 = 1,(76)
since the sum of state projections {P*_w_*_′_(B) = |*w*′(B)〉〈*w*′(B)|} then, amounts to the identity operator: ∑**_*w*′_P*_w_*_′_(B) = 1.

Of interest also is the doubly conditional probability scattering, of the |*w*(A)〉→|*w*′(B)〉 communication in the specified “parameter” state |Ψ〉 of the whole reactive complex. This communication involves the intermediate state |Ψ〉 in the bridge communication [11,47,51,62]:|*w*(A)〉 → |Ψ〉 → |*w*′(B)〉.(77)
Its amplitude can be thus regarded as that of the single-cascade communication determined by the product of two-stage amplitudes,
*A_w_*_(A)→Ψ→*w*′(B)_ = *A_w_*_(A)→Ψ_*A*_Ψ→*w*′(B)_ = 〈*w*(A)|Ψ〉 〈Ψ|*w*′^B^〉 ≡ 〈*w*(A)|P_Ψ_|*w*′(B)〉,(78)
where P_Ψ_ stands for the projection operator onto the “molecular” reference state. The associated conditional probabilities,
*P_w_*_(A)→Ψ→*w*′(B)_ = |*A_w_*_(A)→Ψ→*w*′(B)_|^2^ = *P_w_*_(A)→Ψ_*P*_Ψ→*w*′(B)_ ≡ *P_w_*_(A)→w′(B)_(Ψ),(79)
then, satisfy the intermediate (“bridge”) normalization condition:∑**_*w*′_*P_w_*_(A)→Ψ→*w*′(B)_ = *P_w_*_(A)→Ψ_.(80)

Let us now examine more closely such A→B communications in the molecular state Ψ = Ψ(1, 2) between the stationary states {*ψ_w_*^A^(1)} and {*ψ*_*w*′_^B^(2)} of isolated reactants. They determine the following stage projections:〈*w*(A)|Ψ〉_1_ = *χ_w_*^A^(2)  and  〈Ψ|*w*′(B)〉_2_ = *χ*_*w*′_^B^(1)^*^,(81)
the associated (local) bridge amplitude
*A_w_*_(1)→Ψ(1,2)→*w*′(2)_ = *χ_w_*^A^(2) *χ*_*w*′_^B^(1)^*^,(82)
and resulting density of two-electron probabilities:*P_w_*_(1)→Ψ(1,2)→*w*′(2)_ ≡ *P_w_*_(1)→*w*′(2)_[Ψ(1, 2)] = |*χ_w_*^A^(2)|^2^ |*χ*_*w*′_^B^(1)|^2^.(83)
Their global analogs, measuring probabilities between states rather than locations within states, involve integrations of such local scattering densities over all possible locations of Electron 1 in the network source state, *ψ_w_*^A^(1), and of Electron 2 in the system receiver state, *ψ*_*w*′_^B^(2):*P_w_*_(A)→*w*′(B)_(Ψ) = ∫∫*P_w_*_(1)→*w*′(2)_[Ψ(1, 2)] *d**r***_1_*d**r***_2_                      = [∫|*χ_w_*^A^(2)|^2^*d**r***_2_] [∫|*χ*_*w*′_^B^(1)|^2^*d**r***_1_] = 〈*χ_w_*^A^|*χ_w_*^A^〉_2_ 〈*χ*_*w*′_^B^|*χ*_*w*′_^B^〉_1_              ≡ *ρ_w_*,*_w_*(A) *ρ*_*w*′_,**_*w*′_(B) ≡ *P*(*ψ_w_*^A^|Ψ) *P*(*ψ*_*w*′_^B^|Ψ).(84)
Hence, the stage probabilities in the cascade of Equation (83) read:*P_w(_*_A)→Ψ_ = *P*(*ψ_w_*^A^|Ψ) = 〈*χ_w_*^A^|*χ_w_*^A^〉_2_ = *ρ_w_*_,*w*_(A)  and(85)
*P*_Ψ→*w*′(B)_ = *P*(*ψ*_*w*′_^B^|Ψ) = 〈*χ*_*w*′_^B^|*χ*_*w*′_^B^〉_1_ = *ρ_w_*_′,*w*′_(B).
The double-conditional scattering probabilities between states of isolated reactants indeed observe the bridge normalization of Equation (80).

This *inter*-reactant, external communication system, defined by blocks **P**(A→B) and **P**(B→A) of the conditional probabilities in this resolution of stationary states of isolated reactants, ultimately generates the entropic multiplicities (in bits) of the chemical bonds between both substrates [9,10,11,22,23,24,25,26,27,28,29,30,31,32,62]. For the given molecular state of the whole reactive system, the conditional entropy of the output states, given the input states, ultimately defines the overall IT covalency in the *inter*-reactant bonds, a measure of the information noise in the underlying communication system, while, the complementary descriptor of the overall bond IT ionicity, then, reflects the mutual information in these reactant states, a measure of the information flow between the two subsystems. Elsewhere [101] we have examined the internal communications {**P**(X→X)} in interacting subsystems, which shape electronic structure of the polarized reactants. They have been shown to be determined by the fragment density matrices of Equations (55), (84), and (85).

## 8. Conclusions

Due to Heisenberg’s uncertainty principle of QM, the sharply specified locations of electrons in the position representation of quantum states, which defines the molecular wavefunction and the associated probability distribution in the physical space, precludes the corresponding precise specification of electronic momenta. Therefore, only an effective measure of the latter, consistent with the probability flux definition, is available in quantum description. The current-per-particle measure of the probability velocity, which itself combines the incompatible position and momentum variables of electrons, appears as a natural choice for such an effective local “velocity” descriptor, which gives rise to its vanishing divergence in molecular QM [101]. This simplifies local continuity considerations for the electronic probability “fluid” and separates the “moment”, a “static” aspect of electronic probability density determined by the modulus component of molecular wavefunction, from the “momentum”, a “dynamic” feature of electronic current distribution reflecting the phase gradient of molecular quantum state. To paraphrase Prigogine [102], the former reflects the state (static) electronic structure “of being” while the latter constitutes the its (dynamic) structure “of becoming”. The classical (probability) and nonclassical (current) degrees-of-freedom of molecular states, then respectively determine the system structures “of being” and “of becoming”. Both these patterns carry the information contained in the system (complex) quantum electronic state and contribute to the overall (resultant) entropy and information descriptors.

The distributions of electrons and their current in a molecule determine the classical (modulus) and nonclassical (phase) contributions to the overall information content of the system quantum state. The minimum of the average resultant gradient measure of information, the expectation value of (dimensionless) kinetic energy of electrons, then, establishes the information equilibria in the whole molecular system and its fragments, reflected by the local (“thermodynamic”) phase contribution. The phase aspect of such generalized, phase-transformed equilibrium states is vital for the coherent propagation of electronic communications in molecules. It also distinguishes the bonded (entangled) and nonbonded (disentangled) states of reactants.

In the present analysis, we have reexamined the probability and phase continuities in QM and summarized the resultant measures of the information and entropy content combining the classical and nonclassical contributions. The additive resolution of the wavefunction logarithm have generated the (complex) state continuity relation with the relevant source contribution identified as the (imaginary) phase production. We have also discussed sources of such overall entropy and information descriptors.

The states of isolated and interacting reactants in a simple model of the reactive (donor-acceptor) system have been explored in some detail. We have emphasized the mixed character of electronic states in the entangled (interacting) molecular fragments. Indeed, such subsystems have been shown to be described by their partial density operators, with the quantum expectations of reactant properties determined by the subsystem density matrices for the specified (pure) molecular state. Information principles using the resultant entropy and information measures have also been used to determine phase equilibria [51,52,53,54,55,56,57] in molecular systems and their constituent parts, marking the extreme values of alternative overall measures of electronic entropy (uncertainty, “disorder”) or information (determinicity, “order”) content in electronic wavefunctions. These “thermodynamic” states represent phase-transforms of molecular wavefunctions and generate finite equilibrium currents.

As an illustration, the disentangled (mutually closed) and thermodynamically entangled (mutually open) EO systems of the HZM construction have been examined. In this “plane-wave” type representation the fixed electron densities of molecular fragments generate finite electronic currents due to nonvanishing (local) EO phases, and hence also finite nonclassical contributions to the resultant IT descriptors.

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
