# Peer review of "Information-Theoretic Descriptors of Molecular States and Electronic Communications between Reactants"

_entropy, 2020, doi:10.3390/e22070749_

Round 1

Reviewer 1 Report

This paper uses information theory to discuss chemical bonding, and exploits the phase of the wavefunction to that end. The choice of a time-dependent approach is unconventional and clever.

The weakness of the paper is that it is rather inaccessible; the notation is not always clearly defined, a lot of vernacular is used, and the take-home message for a chemist would like to use the result, without digging through all the mathematics, is unclear. For example, what does the analysis in section 7 say about covalent vs. ionic bonds? About bond strength? Are there simple “take home” formulas that can be used? What are the results of such formulas for simple prototypical chemical systems? Those are my major concerns.

I would like to emphasize that the approach is innovative and correct, and certainly worthy of publication. A few minor points are:

(a) Recent work on the information theory approach should be cited. For example, reviews by the Liu and Ayers groups, and probably some of the references cited therein (and perhaps subsequent work by these authors). [Acta Physico-Chimica Sinica 32, 98 (2016); J. Phys. Chem. 122, 4219 (2018)]
(b) Several typos. E.g., in line 310 “originating” is misspelled. Presumably, the editor will fix these before publication.

Author Response

Extracting the physical content of intuitive chemical concepts often require

complicated mathematical framework of molecular quantum mechanics, which could  make such papers inaccessible for ordinary chemists. One has to accept it.

The rudiments of communication theory of the chemical bond have been described in both scientific papers and monographs (quoted in References), and there is no need to repeat them. However, I have added some new comments in the rewritten Introduction, which should make the paper comprehension easier. The refrences for this resultant IT analysis have been carefully planned and there is no need for any etension. Not all papers using Shannon's entropy or Fisher information are essential in this context.  

Reviewer 2 Report

This paper explores molecular binding from the “communication theory” viewpoint. The innovation, based on my limited knowledge of the field, is in the use of a time-dependent framework. This seems counterintuitive at first, since the ground state (indeed, any stationary state) of a system is not time-dependent, but the author makes good use of the fact that two (sub)systems, when their wavefunctions are melded together, do not give a eigenstate of the supersystem. (I am less convinced by the argument using equi-density orbitals, which are indeed a (very) poor and nonstationary wavefunction, but not one I would expect to learn anything from.)

The paper is unquestionably novel and innovative and should be published after minor changes, denoted below.

Specific concerns I have are below:

  1. Lines 25-29 . The introduction is heavily biased towards the authors own work, and does not cite other key references. For example, there are recent reviews on information theory in chemistry from the Ayers and Liu groups,1-6 and a somewhat older review-ish article from the Geerlings group.7 There are also many other contributions related to conceptual DFT, e.g. 6, 8-14.
  2. Line 49. There is significant recent work, from a different but nonetheless information-theoretic perspective, on the entangled/disentangled nature of chemical binding.15, 16 The analysis seems, to a novice like me, to be not dissimilar to that in section 7. Perhaps the author can comment on the similarities/differences?
  3. Line 52. There was a lot of work on information theory in the “general reactivity” context from Levine.17-20
  4. Line 123. Typo. “seen to
  5. Lines 209, 210. Does the author really mean to denote the acid and base as cations? I would assume the acid might be a cation, but that the base should be neutral or anionic. I think this is probably not what is meant (based on Eq. (43)) but what is meant is obscure. Also the use of R in Eq. (43) to mean a system, with the use of R in preceding section (cf. Eq. (17)) to denote the real part of the wavefunction, together with the use of R in line 212 to denote bond length…it is very confusing!
  6. In line 216, the “+” that was in Eq. (43) has been changed to a “*” with no explanation of notation.
  7. Line 220. typo. “of the”
  8. Line 310. typo. “originating”
  9. The analysis in the appendix, far from being an afterthought, is one of the most interesting and clear results in the paper. I would not relegate it to the appendix.
  10. Heidar-Zadeh, F.;  Ayers, P. W.;  Verstraelen, T.;  Vinogradov, I.;  Vohringer-Martinez, E.; Bultinck, P., Information-Theoretic Approaches to Atoms-in-Molecules: Hirshfeld Family of Partitioning Schemes. Journal of Physical Chemistry A 2018, 122 (17), 4219-4245.
  11. Zhou, X. Y.;  Rong, C. Y.;  Lu, T.;  Zhou, P. P.; Liu, S. B., Information Functional Theory: Electronic Properties as Functionals of Information for Atoms and Molecules. Journal of Physical Chemistry A 2016, 120 (20), 3634-3642.
  12. Liu, S.-B., Information-Theoretic Approach in Density Functional Reactivity Theory. Acta Physico-Chimica Sinica 2016, 32 (1), 98-118.
  13. Rong, C. Y.;  Lu, T.;  Ayers, P. W.;  Chattaraj, P. K.; Liu, S. B., Scaling properties of information-theoretic quantities in density functional reactivity theory. Physical Chemistry Chemical Physics 2015, 17 (7), 4977-4988.
  14. Liu, S.;  Rong, C.;  Wu, Z.; Lu, T., Renyi Entropy, Tsallis Entropy and Onicescu Information Energy in Density Functional Reactivity Theory. Acta Physico-Chimica Sinica 2015, 31 (11), 2057-2063.
  15. Liu, S.;  Rong, C.; Lu, T., Information Conservation Principle Determines Electrophilicity, Nucleophilicity, and Regioselectivity. Journal of Physical Chemistry A 2014, 118 (20), 3698-3704.
  16. De Proft, F.;  Ayers, P. W.;  Sen, K. D.; Geerlings, P., On the importance of the "density per particle" (shape function) in the density functional theory. Journal of Chemical Physics 2004, 120, 9969-9973.
  17. Heidar-Zadeh, F.;  Fuentealba, P.;  Cardenas, C.; Ayers, P. W., An information-theoretic resolution of the ambiguity in the local hardness. Physical Chemistry Chemical Physics 2014, 16 (13), 6019-6026.
  18. Yu, D. H.;  Rong, C. Y.;  Lu, T.;  De Proft, F.; Liu, S. B., Aromaticity Study of Benzene-Fused Fulvene Derivatives Using the Information-Theoretic Approach in Density Functional Reactivity Theory. Acta Physico-Chimica Sinica 2018, 34 (6), 639-649.
  19. Cao, X. F.;  Rong, C. Y.;  Zhong, A. G.;  Lu, T.; Liu, S. B., Molecular acidity: An accurate description with information-theoretic approach in density functional reactivity theory. Journal of Computational Chemistry 2017, 39 (2), 117-129.
  20. Wu, Z. M.;  Rong, C. Y.;  Lu, T.;  Ayers, P. W.; Liu, S. B., Density functional reactivity theory study of S(N)2 reactions from the information-theoretic perspective. Physical Chemistry Chemical Physics 2015, 17 (40), 27052-27061.
  21. Tsirelson, V. G.;  Stash, A. I.; Liu, S. B., Quantifying steric effect with experimental electron density. Journal of Chemical Physics 2010, 133, 114110.
  22. Liu, S. B.;  Hu, H.; Pederset, L. G., Steric, Quantum, and Electrostatic Effects on S(N)2 Reaction Barriers in Gas Phase. Journal of Physical Chemistry A 2010, 114, 5913-5918.
  23. Nagy, A.; Liu, S. B., Local wave-vector, Shannon and Fisher information. Physics Letters A 2008, 372, 1654-1656.
  24. Szalay, S.;  Barcza, G.;  Szilvasi, T.;  Veis, L.; Legeza, O., The correlation theory of the chemical bond. Scientific Reports 2017, 7.
  25. Duperrouzel, C.;  Tecmer, P.;  Boguslawski, K.;  Barcza, G.;  Legeza, O.; Ayers, P. W., A quantum informational approach for dissecting chemical reactions. Chemical Physics Letters 2015, 621, 160-164.
  26. Levine, R. D.; Bernstein, R. B., Energy consumption and energy disposal in elementary chemical reactions: The information theoretic approach. Accounts of Chemical Research 1974, 7, 393.
  27. Kaplan, H.;  Levine, R. D.; Manz, J., Microscopic reversibility and probability matrices for molecular collisions -Information theoretic synthesis. Molecular Physics 1976, 31 (6), 1765-1782.
  28. Levine, R. D., Information-theory approach to molecular reaction dynamics. Annual Review of Physical Chemistry 1978, 29, 59-92.
  29. Levine, R. D., Molecular reaction dynamics. Cambridge UP: Cambridge, 2005.

Author Response

The present Quantum Information Theoretic paper is not a review, so the list of references has been limited only to an essential selection. I do not find the references listed by the reviewer essential in this context.